# Low Efficacy of Genetic Tests for the Diagnosis of Primary Lymphedema Prompts Novel Insights into the Underlying Molecular Pathways

**DOI:** 10.3390/ijms23137414

**Published:** 2022-07-03

**Authors:** Gabriele Bonetti, Stefano Paolacci, Michele Samaja, Paolo Enrico Maltese, Sandro Michelini, Serena Michelini, Silvia Michelini, Maurizio Ricci, Marina Cestari, Astrit Dautaj, Maria Chiara Medori, Matteo Bertelli

**Affiliations:** 1MAGI’s LAB, 38068 Rovereto, Italy; stefano.paolacci@assomagi.org (S.P.); paolo.maltese@assomagi.org (P.E.M.); astrit.dautaj@assomagi.org (A.D.); chiara.medori@assomagi.org (M.C.M.); matteo.bertelli@assomagi.org (M.B.); 2MAGI Group, 25010 San Felice del Benaco, Italy; michele.samaja@assomagi.org; 3Vascular Diagnostics and Rehabilitation Service, Marino Hospital, ASL Roma 6, 00047 Marino, Italy; sandro.michelini@aslroma6.it; 4Unit of Physical Medicine, “Sapienza” University of Rome, 00185 Rome, Italy; serenamichelini@gmail.com; 5Neurosurgery, University of Tor Vergata, 00133 Rome, Italy; silviamichelini1992@gmail.com; 6Division of Rehabilitation Medicine, Azienda Ospedaliero-Universitaria, Ospedali Riuniti di Ancona, 60126 Ancona, Italy; maurizio.ricci@ospedaliriuniti.marche.it; 7Study Centre Pianeta Linfedema, 05100 Terni, Italy; cestari.marina@libero.it; 8Lymphology Sector of the Rehabilitation Service, USLUmbria2, 05100 Terni, Italy; 9MAGI Euregio, 39100 Bolzano, Italy

**Keywords:** lymphedema, molecular pathways, genetic screening, diagnostic genes, candidate genes, PI3K/AKT, RAS/MAPK, Rho/ROCK

## Abstract

Lymphedema is a chronic inflammatory disorder caused by ineffective fluid uptake by the lymphatic system, with effects mainly on the lower limbs. Lymphedema is either primary, when caused by genetic mutations, or secondary, when it follows injury, infection, or surgery. In this study, we aim to assess to what extent the current genetic tests detect genetic variants of lymphedema, and to identify the major molecular pathways that underlie this rather unknown disease. We recruited 147 individuals with a clinical diagnosis of primary lymphedema and used established genetic tests on their blood or saliva specimens. Only 11 of these were positive, while other probands were either negative (63) or inconclusive (73). The low efficacy of such tests calls for greater insight into the underlying mechanisms to increase accuracy. For this purpose, we built a molecular pathways diagram based on a literature analysis (OMIM, Kegg, PubMed, Scopus) of candidate and diagnostic genes. The PI3K/AKT and the RAS/MAPK pathways emerged as primary candidates responsible for lymphedema diagnosis, while the Rho/ROCK pathway appeared less critical. The results of this study suggest the most important pathways involved in the pathogenesis of lymphedema, and outline the most promising diagnostic and candidate genes to diagnose this disease.

## 1. Introduction

A chronic inflammatory disorder caused by abnormal accumulation of interstitial fluid resulting from ineffective flow and fluid uptake by the lymphatic system, lymphedema affects not only the lower limbs, but also the upper limbs, genitals, and face. By evolving in irreversible fibrosis, lymphatic damage, and tissue hypertrophy, it causes thickening and hardening of the limb [1,2,3]. Lymphedema may be classified as primary when caused by genetic mutations with typical Mendelian inheritance, or secondary when it is acquired following injury, infection, or surgery. The prevalence of primary and secondary lymphedema is 1/100,000 and 1/1000, respectively. Overall, lymphedema affects 140–250 million people worldwide [4,5,6].

The aims of this study are twofold. First, we aim to assess to what extent the current genetic tests can detect causative mutations in primary lymphedema patients. Second, we aim to identify the candidate molecular pathways for lymphedema pathogenesis. At present only a few studies have focused on the pathways underlying this disease, which remain largely unknown. For these purposes, we used an up-to-date technique to detect genetic variants and an electronic search to build several molecular pathway diagrams, in order to define new diagnostic and therapeutic targets for lymphedema.

## 2. Results

### 2.1. Efficacy of the Current Genetic Tests

Caucasian lymphedema patients (*n* = 147) were recruited in clinics in the Italian regions of Campania, Latium, Lombardy, Marches, Puglia, Umbria, Veneto, and Trentino-Alto Adige. The recruited patients underwent several clinical investigations from 2019 to 2021 to diagnose primary lymphedema and exclude secondary causes of the pathology. The diagnosis of lymphedema was confirmed as reported in previous articles [7,8]. Briefly, lymphedema was diagnosed via three-phase lymphoscintigraphy according to the protocol of Bourgeois. To exclude secondary lymphedema patients, specific investigation and/or biochemical tests were performed, considering medical history and physical examination. All patients received genetic counseling to explain the risks and benefits of genetic testing. Figure 1 reports the results of the genetic tests of the probands recruited for this study and analyzed for the genes, reported in Appendix A. When using the classification described in Materials and Methods, 63 probands had negative results, 73 had inconclusive results, and 11 has positive results. Mutations in the genes CBL, CELSR1, FAT4, FLT4, HGF, RIT1, and VEGFC were identified in the patients who tested positive in the genetic test. The low efficacy of the established genetic tests clearly calls for greater insight into the underlying mechanisms in order to meet the need for more accurate results.

### 2.2. Study Cohort Characteristics

Table 1 reports the clinical characteristics of the probands analyzed for this study.

### 2.3. Molecular Pathways Involved in Primary Lymphedema

To meet the second aim of this study, we performed a search based on the diagnostic and candidate genes, as explained in the Materials and Methods section.

#### 2.3.1. Diagnostic Gene Pathways

We considered thirty-five genes to be correlated with several forms of primary lymphedema, following the criteria presented in Materials and Methods. The Appendix A reports the list of these genes along with their gene–phenotype relationship and the pathways in which they are involved. Figure 2 reports the molecular pathways that encompass the proteins encoded by the mentioned genes in a somatic cell. The majority of these proteins appear to participate to the RAS/MAPK (violet) and the PI3K/AKT (lilac) pathways, thereby activating several transcription factors (orange). Remarkably, some of them are involved in the VEGF-C/VEGFR-3 (blue) and in the HGF/MET (red) signaling cascades. The light blue proteins are those whose outcome is known, i.e., lymphatic valve formation, but the molecular pathways are uncertain.

#### 2.3.2. Candidate Gene Pathways

We considered seventy-one genes as candidate genes, following the criteria presented in Materials and Methods. Appendix A reports a list of these genes along with their gene–phenotype relationship and the pathways in which they are involved. Figure 3, Figure 4, Figure 5 and Figure 6 report the interactions of the candidate genes with four pathways:PI3K/AKT pathway (Figure 3);RAS/MAPK pathway (Figure 4);Rho/ROCK pathway (Figure 5);Secondary pathways (Figure 6).

## 3. Discussion

This study shows that the efficacy of genetic tests performed in a cohort of Italian lymphedema patients enables correct diagnosis in only 7% of the tested patients, which means that the majority of such patients do not have a specific and reliable genetic test for this disease. Despite past studies having attempted to relate genetic findings with key physiological or cellular mechanisms [9,10,11], current knowledge of the genetic basis of lymphedema is clearly insufficient to explain the majority of primary lymphedema cases. Thus, in this study, we propose candidate molecular pathways in the search for new genetic targets for the diagnosis of primary lymphedema and for developing new therapeutic strategies.

### 3.1. Molecular Pathways Involved in Primary Lymphedema

#### 3.1.1. VEGF-C/VEGFR-3 Pathway

The most important pathway involved in lymphedema is probably the VEGF-C/VEGFR-3 pathway described in Figure 2 [12]. The binding of VEGF-C to its receptor controls lymphangiogenesis, and mutations in several genes involved in this signal transduction cascade result in different forms of lymphedema. First, heterozygous loss-of-function mutations in the VEGF-C [13] and FLT4 (VEGFR-3) [14,15] genes are correlated with, respectively, lymphatic malformation 4 (OMIM: 6115907) and lymphatic malformation 1 (OMIM: 153100). VEGF-C is expressed where the first lymph sacs develop during embryogenesis and in regions of lymph-vessel sprouting [16]. Its importance during embryogenesis is supported by in vivo studies: mice deleted for Vegfc do not develop lymphatic vessels, while the development of lymph vasculature is delayed in Vegfc heterozygous mice, sometimes leading to chylous ascites formation in pups [17,18]. VEGFR-3 is expressed almost exclusively in the lymphatic endothelium of human adult tissues. Its activation decreases apoptosis and increases the proliferation and migration of endothelial cells in vitro, while its deletion disrupts vascular formation and haematopoiesis in mice, resulting in cardiovascular failure and embryonic death [16,19,20].

Apart from VEGF-C and VEGFR-3, many other proteins participate in this pathway, such as CCBE1, ADAMTS3, and PTPN14. CCBE1 is able to enhance in vivo sprouting of lymphatic vessels. A lack of it impairs VEGF-C and VEGFR-3 Erk signaling [21], while its homozygous and compound heterozygous loss-of-function mutations [22,23] cause Hennekam lymphangiectasia-lymphedema syndrome 1 (OMIM: 235510). ADAMTS3 triggers VEGFR-3 signaling [24]. Its compound heterozygous loss-of-function mutations [24] cause Hennekam lymphangiectasia-lymphedema syndrome 3 (OMIM: 6118154). Finally, homozygous loss-of-function mutations [25,26] to PTPN14 cause choanal atresia and lymphedema (OMIM: 613611). PTPN14 interacts with VEGFR-3, and its deletion in mice causes lymphatic hyperplasia with lymphedema [25]. VEGF-C and VEGFR-3 binding activates the RAS/MAPK and the PI3K/AKT pathways.

#### 3.1.2. HGF/MET Pathway

HGF/MET signaling influences several biological activities, including differentiation, cell motility, growth, and survival [27]. Moreover, HGF exhibits lymphangiogenic activity, stimulating the proliferation of LECs [28]. Mutations of both proteins are correlated with the development of several pathologies, among which are primary lymphedema, lymphedema/lymphangiectasia, and breast cancer-associated secondary lymphedema [8,27] (Figure 2). Moreover, CBL and PTPN11 also participate in the initial phases of the HGF/MET signaling pathway. CBL regulates MET internalization, downregulating its signaling; heterozygous loss-of-function mutations [29] in CBL cause Noonan syndrome-like disorder (OMIM: 613563). PTPN11, also known as SHP2, participates in MET’s signal transduction, activating both RAS and AKT, thus acting on both the AKT/PI3K pathway and the RAS/MAPK pathway. PTPN11 heterozygous gain-of-function mutations [30,31] or duplications [32,33] cause Noonan syndrome 1 (OMIM: 163950).

#### 3.1.3. PI3K/AKT Pathway

Activated by both VEGF-C/VEGFR-3 and HGF/MET signaling, the PI3K/AKT pathway is an essential component of the process of lymphangiogenesis, stimulating the survival of LECs [12] (Figure 2). The relevance of the PI3K/AKT pathway in the etiopathogenesis of lymphedema is underlined by several studies that couple gain-of-function mutations in AKT1 [34] and PI3KA [35,36,37] with syndromic lymphatic malformations. The PI3K/AKT pathway starts with the activation of PI3KA by VEGFR-3, which, in turn, activates AKT. PI3KA can also be activated by RAS, linking the RAS/MAPK and the PI3K/AKT pathways [38]. Mosaic gain-of-function mutations [34] in AKT1 cause Proteus syndrome (OMIM: 179620), while somatic gain-of-function mutations [35,36,37] in PI3KA cause CLAPO (OMIM: 613089) and CLOVE (OMIM: 612918) syndromes. AKT is able to activate both mTOR and IKBKG. The former stimulates cell survival and proliferation [39], while the latter activates NFkB [12]. Finally, NFkB stimulates PROX1 and VEGFR-3 transcription, increasing LECs’ responsiveness to VEGF-C and VEGF-D and initiating lymphangiogenesis [40,41]. The PI3K/AKT pathway can be activated by two accessory proteins: KIF11, which activates PI3KA, and PIEZO1, which activates AKT, increasing Ca^2+^ cytosolic concentration. KIF11 heterozygous nonsense mutations [42] cause microcephaly with or without chorioretinopathy, lymphedema, or mental retardation (OMIM: 152950), while PIEZO1 homozygous or compound heterozygous loss-of-function mutations [43] cause lymphatic malformations 6 (OMIM: 616843).

Several proteins encoded by the proposed candidate genes influences the activation of the PI3K/AKT pathway (Figure 3). In particular, many receptors activate the PI3K/AKT pathway upon binding to their ligand (CALCRL and ADM, LPAR and LPA, MET and HGF, NRP and ANGPTL, S1P and S1PR1, TIE and ANGPT, and VEGFR and VEGF). Apart from these, the cytoskeleton remodeling initiated by the Rho/ROCK pathway activates integrin and its effectors FAK (SYR2) and SRC. They activate the PI3K/AKT and NFkB pathways [44,45], which finally trigger transcription factors such as FOXC1 and repress apoptosis, inhibiting PPP1R13B activity. Other candidate genes that participate in the PI3K/AKT pathway are ARAP3, SYK, RELN, and SVEP1 (see Appendix A for references). 

#### 3.1.4. RAS/MAPK Pathway

The RAS/MAPK pathway plays a pivotal role in promoting lymphangiogenesis [38]. Mutations in different genes involved in this pathway are correlated with primary lymphedema onset (Figure 2). Both VEGF-C/VEGFR-3 and HGF/MET signaling activate the RAS/MAPK pathway by acting on GRB2, which, in turn, activates SOS1. SOS1 heterozygous [46] gain-of-function [47,48] mutations are correlated with Noonan syndrome 4 (OMIM: 610733). SOS1 activates RIT1 and RAS. RIT1 heterozygous gain-of-function mutations [49,50] causes Noonan syndrome 8 (OMIM: 615355). As for RAS, human cells express this protein in three main isoforms: NRAS, KRAS, HRAS [51]. Mutation in any of them causes several disorders correlated with lymphedema: KRAS heterozygous gain-of-function mutations [52,53] result in Noonan syndrome 3 (OMIM: 609942); NRAS heterozygous gain-of-function mutations [54] result in Noonan syndrome 6 (OMIM: 613224); and HRAS heterozygous gain-of-function mutations [55,56] result in Costello syndrome (OMIM: 218040). Going further into the signaling cascade, both RIT1 and RAS activate BRAF, heterozygous gain-of-function mutations [57] which result in Noonan syndrome 7 (OMIM: 613706). BRAF activates MEK, which, in turn, activates ERK, which finally stimulates proliferation and migration, in the same way as it also stimulates PROX1 expression [38].

Several proteins encoded by the candidate genes participate in the RAS/MAPK pathway (Figure 4). Many receptors activate this pathway upon binding to their ligand (CALCRL and ADM, LPAR and LPA, MET and HGF, S1P and S1PR1, TIE and ANGPT, and VEGFR and VEGF). Its signaling cascade, in which several other candidate proteins are involved (ITGA5/9, MAP4K4, SYK, SPRED1, LZTR1, and PPPCB1), ends the activation of ERK, which triggers many transcription factors (FOXC1, NFATC1, PROX1).

#### 3.1.5. Rho/ROCK Pathway

Proteins encoded by the proposed candidate genes are involved in the Rho/ROCK pathway (Figure 5). Although mutations in the proteins of this pathway are not correlated with lymphedema in OMIM, several studies suggest its importance in many physiological processes correlated with this pathology. Indeed, RhoA was proven to have a pivotal role in cell migration and motility [58,59], probably acting on cytoskeleton remodeling of endothelial cells [60,61]. The promotion of cellular migration enhances angiogenesis and lymphangiogenesis, as well as tumor lymphatic metastasis [62,63,64]. Lymphangiogenesis could also be linked to Rho interaction with VANGL2, a key regulator of mechanotransduction and the planar-cell-polarity pathway [65]. As for the mechanism of action, several studies suggest that RhoA can be activated by different proteins encoded by the proposed candidate genes (ANGPTL4, CDH5, LPAR, PDPN, PLXNA1, S1PR, and SEMA3A), and it then activates its effector ROCK. ROCK initiates a signaling cascade involving MLC phosphorylation, stress fiber contraction, non-receptor tyrosine kinase activation, and finally, the activation of the Akt and Erk pathways [45,59]. Rho and ROCK participate in the correct activation of lymphatic pumps, participating in the active transport of lymph [66]. Disfunction in lymph transport leads to several pathologies, among which the most common is lymphedema [67].

#### 3.1.6. Transcription Factors

Many transcription factors are involved in the function of the lymphatic system. Their mutations are linked to several diseases (Figure 2). In particular, mutations in SOX18, PROX1, GATA2, and FOXC2 cause various forms of syndromic lymphedema. Indeed, either heterozygous or homozygous loss-of-function mutations [68,69,70] in SOX18 cause hypotrichosis-lymphedema-telangiectasia syndrome (OMIM: 607823); heterozygous loss-of-function mutations [71] in GATA2 cause Emberger syndrome (OMIM: 614038); and heterozygous gain-of-function mutations [7,72,73] in FOXC2 cause Lymphedema-distichiasis syndrome (OMIM: 153400). Moreover, although mutations in PROX1 are not directly linked to lymphedema in OMIM, several studies support its essential role in lymphangiogenesis [12,38,40]. Two PROX1 variants were recently identified in lymphedema patients [74]. Indeed, it upregulates the transcription of VEGFR3, stimulating the RAS/MAPK and the AKT/PI3K pathways. The expression of PROX1 is regulated by SOX18 and GATA2, while the latter also controls the expression of FOXC2 [12]. FOXC2, a member of the forkhead-box (FOX) family, regulates the development of several systems during embryogenesis, particularly the lymphatic and blood vascular system [75,76].

Apart from the diagnostic genes, several candidate genes code for transcription factors or influence their activity (Figure 6). For example, they can act as transcriptional activators of PROX1, such as NR2F2, SOX17, and HHEX. Among them, NR2F2 is negatively regulated by NOTCH signaling through HEY2 expression [77]. Moreover, they can activate FOXC2, such as CDK5 and FOXC2-AS1 [75,78]. PROX1 and FOXC2 expression is also increased by CHD5 through the β-catenin pathway, regulating lymphatic vessels and valve development [79,80]. Other candidate genes are correlated with SOX18 expression: VCAM1 expression is increased by SOX18, while GDF2 signaling reduces SOX18 expression [81,82]. RORC and SMARCA4 influence LECs’ fate determination [83], while HOX10 is proven to reduce RhoC expression, influencing cytoskeletal organization [84]. Finally, ERK and AKT activity can stimulate different transcription factors, among which are PROX1 and NFATC1 (stimulated by ERK) and FOXC1 (stimulated by both AKT and EKR) [38,85].

### 3.2. Physiological Outcomes

#### 3.2.1. Planar Cell Polarity and Lymphatic Valve Development

The control of planar cell polarity (PCP) has a pivotal role in the function of the lymphatic system, with the relative mutations in genes involved in this pathway correlated with the development of primary lymphedema [67,86]. In particular, heterozygous loss-of-function mutations [67,87,88] in CELSR1 are correlated with Lymphatic malformation 9 (OMIM: 619319), while homozygous mutations [89] in FAT4 are correlated with Hennekam lymphangiectasia-lymphedema syndrome 2 (OMIM: 616006). CELSR1 and FAT4 control planar cell polarity in LECs, participating in the morphogenesis of lymphatic valves [67,90]. Furthermore, the connexins GJC2 and GJA1 were proven, in vivo, to be involved in lymphatic valve development [86,91]. Heterozygous loss-of-function [92] mutations in the former are linked to the development of Lymphatic malformation 3 (OMIM: 613480), while heterozygous and compound heterozygous loss-of-function mutations [93] in the latter are linked to oculodentodigital dysplasia (OMIM: 164200). Finally, PIEZO1 was proven, in vivo, to be correlated with lymphatic valve development [94]. Unidirectional valves participate in lymph transport: they counteract lymph reflux during the contraction of lymphangions, and their disrupted development reduces lymph drainage, resulting in lymphedema [91,95].

Several candidate genes are involved in the correct development of lymphatic valves. They can influence the PCP pathway (VANGL2 and SDC4), and the hippo pathway (DCHS1 and PPP1CB), or they can transduce mechanical tension in LECs (PECAM1 and CDH5) [96,97,98,99]. Mechanical forces are proven to contribute to endothelial cell fate commitment, influencing LECs’ shape and alignment. They promote sprouting, development, and maturation of the lymphatic network, and they coordinate lymphatic valve morphogenesis [99,100]. Many of the proposed candidate genes are linked to lymphatic valve morphogenesis (EFNB2, EMILIN1, GJA4, ITGA9, NFATC1, PECAM1, and PLXNA1) (see Appendix A). Last, but not least, CYP26B1, a gene involved in retinoic acid metabolism, is proven to participate in lymphangiogenesis and lymphatic valve development [101,102].

#### 3.2.2. Lymph Pumping

Apart from lymphatic valves, lymph transport is enabled by active pumping, which stems from the rhythmic contraction of vascular smooth cells. The contraction of the lymphatic pumping unit, the lymphangion, enables the efficient pumping of lymph through lymphatic vessels up to the central veins. The malfunctions of lymph pumping affect in vivo and ex vivo lymph transport, causing lymphedema-related disorders [87,103]. Different proteins are proven to be involved in correct lymph transport, among which are the ones encoded by ten of our candidate genes (ADM, ARAF, CALCRL, MAP2K1, MAP2K2, NPPA, NPPB, PPP1CB, RAF1, and RAMP2) (see Appendix A).

#### 3.2.3. Macrophage Activation and Lymphangiogenesis

SVEP1 and ACKR2 are involved in processes related to cell adhesion or macrophage recruitment to LECs. SVEP1 is a large extracellular mosaic protein with functions in protein interactions and adhesion, and its mutations have been found by a recent study in lymphedema patients [104]. Furthermore, the chemokine-scavenging receptor ACKR2 is proven to influence lymphangiogenesis; it regulates lymphatic vessel density, acting on the recruitment of pro-lymphangiogenic macrophages [105]. Macrophages seem to be influenced by another candidate gene, FABP4. Its gene expression levels were correlated with CLEC10A, a marker of alternative macrophage polarization [106]. Other candidate genes (IKBKG, PPP1R13B, S1PR1, S1PR2, and VCAM1) participate in the NFkB pathway (see Appendix A), which is another relevant pathway in inflammation and lymphangiogenesis [40].

### 3.3. Limitations and Future Perspectives

The major limitation in this study is the lack of information for some diagnostic and candidate genes in the literature on their molecular activity. Thus, for this study, we relied on published papers that evaluate their cellular or tissue effects, and so, their physiological outcomes. Moreover, for many genes, molecular pathway data in LECs were unavailable; therefore, we relied on studies performed on different cell types. Finally, the provided list of genes should not be considered exhaustive, considering that further research could discover other possibly important genes and pathways involved in lymphedema.

The first diagnosis of lymphedema is usually based on clinical symptoms, but many symptoms of this pathology are subclinical in its first stages. Thus, the molecular diagnosis of lymphedema could be useful in the scope of preventive medicine, and focusing our research on the molecular activity of diagnostic and candidate genes will permit the definition of new diagnostic and therapeutic targets.

Ceasing to consider lymphedema only as a monogenic disease could be useful to explore new possibilities of diagnosis and treatment of this pathology still too much left aside. Lymphedema is now considered a Mendelian disorder, with variants of single genes being at the basis of its etiopathogenesis, but it could also arise from multigenic effects and from post-zygotic mutations. In multigenic diseases, polymorphisms of different genes, together with external factors, participate in the development of the pathology. In order to define the roles of specific genes in genetic disorders, the main strategy is to analyze their association with the disease in families with several affected individuals [107]. Performing segregation analyses could also be helpful for the analysis of individuals firstly diagnosed as negative. Therefore, association studies and new knowledge in this field could be a new way forward to develop personalized medicine treatments for lymphedema patients.

## 4. Materials and Methods

### 4.1. Genetic and Data Analysis

Approximately 5 mL of either peripheral blood or saliva [108] from probands were used for DNA extraction using a commercial kit (Blood DNA kit E.N.Z.A., Omega Bio-tek, Inc., Doraville, GA, USA or Exgene Clinic SV mini, GeneAll Biotechnology, Seoul, South Korea). PCR and direct sequencing of amplified fragments (Thermo PX2 thermocycler, Beckmann Coulter CEQ 8000 sequencer) were used to analyze exons coding for the studied genes, and the respective portions of the intron regions adjacent to the exons. Primer sequences, PCR reaction conditions, and sequencing conditions are available on request. All the samples were then analyzed using NGS Sequencing, and poorly covered target regions were confirmed using Sanger sequencing, as reported [8].

We classified the genetic test as negative, inconclusive, or positive following the guidelines of the American College of Medical Genetics and Genomics [109], which set criteria for identifying clinically relevant sequence variants. In brief, a genetic test was classified as:Negative when no variants were found in the analyzed genes;Inconclusive when the detected variants could be causative of the disease, but their effect is uncertain;Positive when the detected variants were correlated with the disease.

Appendix A reports the diagnostic genes. Diagnostic genes were those that correlated with lymphedema, with lymphatic malformations, or with syndromic forms of lymphedema as from OMIM [110], Kegg [111] or scientific literature until 2019, when the genetic testing of patients began.

### 4.2. Creation of Molecular Pathway Diagrams

Appendix A reports the candidate genes. Candidate genes were identified in 2019 following the criteria established by [112]. In brief, a gene is considered a candidate gene if it meets at least one of these criteria:The gene is involved in the development of the lymphatic system in vivo;Gene mutations cause in vivo lymphatic defects;The gene participates in a molecular pathway that is important in lymphedema pathogenesis.

We searched for each of the identified genes in OMIM, Kegg, PubMed [113] and Scopus [114], to build a database based on the published articles that report detailed characterizations of the molecular pathways in in vitro and in vivo models. For in vitro studies, preference was given to data obtained from lymphatic endothelial cells (LECs). PathVisio software was used for the creation of molecular pathway diagrams [115]. Of the 92 studies used for the creation of Appendix A, 65 were not cited in the main article [116,117,118,119,120,121,122,123,124,125,126,127,128,129,130,131,132,133,134,135,136,137,138,139,140,141,142,143,144,145,146,147,148,149,150,151,152,153,154,155,156,157,158,159,160,161,162,163,164,165,166,167,168,169,170,171,172,173,174,175,176,177,178,179,180].

## 5. Conclusions

The low efficacy of established genetic tests for lymphedema calls for a deeper insight into the underlying molecular mechanisms, in order to define new diagnostic and therapeutic targets. In this study, we identified a few molecular paths that may explain the link between the observed genetic alterations and the physiological outcomes of lymphedema. The most important pathways involved in the pathogenesis of lymphedema are the PI3K/AKT and the RAS/MAPK pathways, while the Rho/ROCK pathway and other secondary pathways appeared less critical.

## Figures and Tables

**Figure 1 ijms-23-07414-f001:**
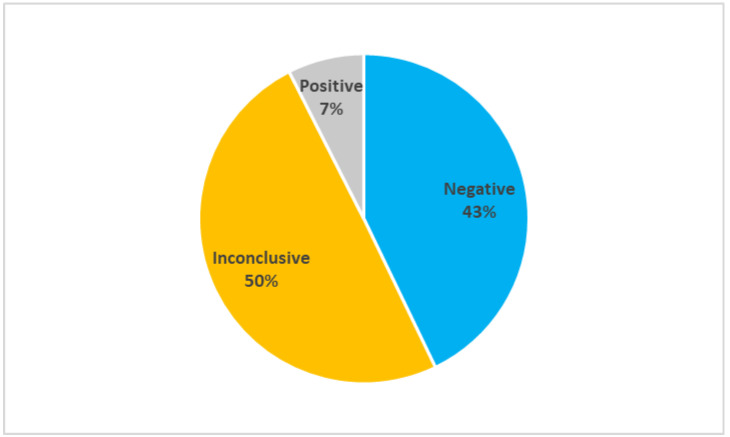
Results of genetic tests (*n* = 147). The pie reports the percentage of negative, inconclusive, and positive test on the whole population considered.

**Figure 2 ijms-23-07414-f002:**
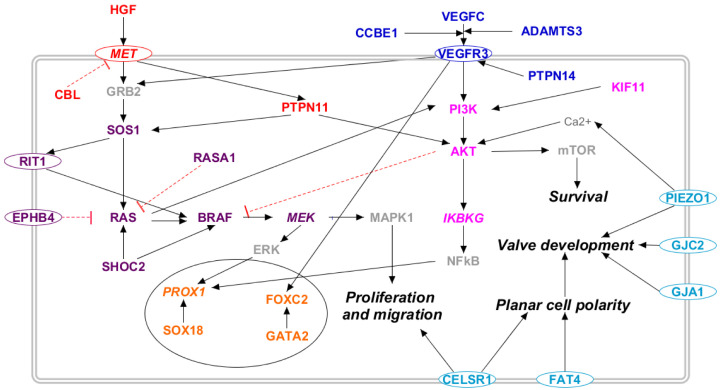
Molecular pathways involved in primary lymphedema. The 28 proteins coded by the genes related to primary lymphedema are placed along known molecular pathways in a somatic cell: RAS/MAPK (violet), PI3K/AKT (lilac), VEGF-C/VEGFR-3 (blue), and HGF/MET (red). The transcription factors are marked in orange, and the light blue proteins are those whose outcome is known, i.e., altered lymphatic valve formation, but the molecular pathway is uncertain. Proteins not encoded by either diagnostic or candidate genes are marked in grey. Proteins encoded by candidate genes are reported in italic. When circled, the protein is a membrane protein. Black arrows represent a positive interaction, while red T-bars represent an inhibition.

**Figure 3 ijms-23-07414-f003:**
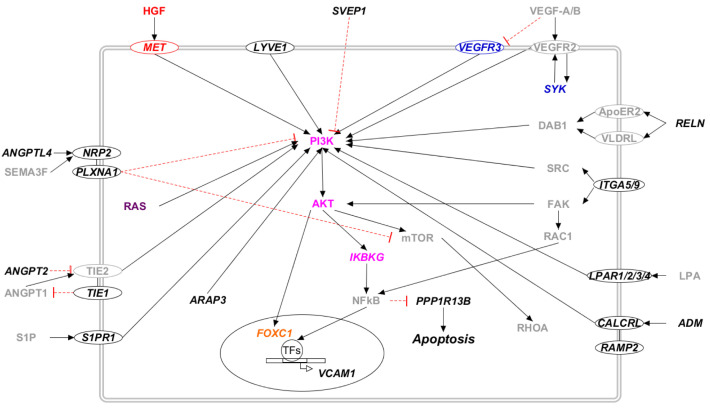
The PI3K/AKT pathway in candidate genes for primary lymphedema. Proteins that participate in one of the previously reported molecular pathways are colored: RAS/MAPK (violet), PI3K/AKT (lilac), VEGF-C/VEGFR-3 (blue), and HGF/MET (red). The transcription factors are marked in orange. Proteins that do not participate in any of the previously reported pathways are marked in black. Proteins not encoded by either diagnostic or candidate genes are marked in grey. Proteins encoded by candidate genes are reported in italic. When circled, the protein is a membrane protein. Black arrows represent a positive interaction, while red T-bars represent an inhibition.

**Figure 4 ijms-23-07414-f004:**
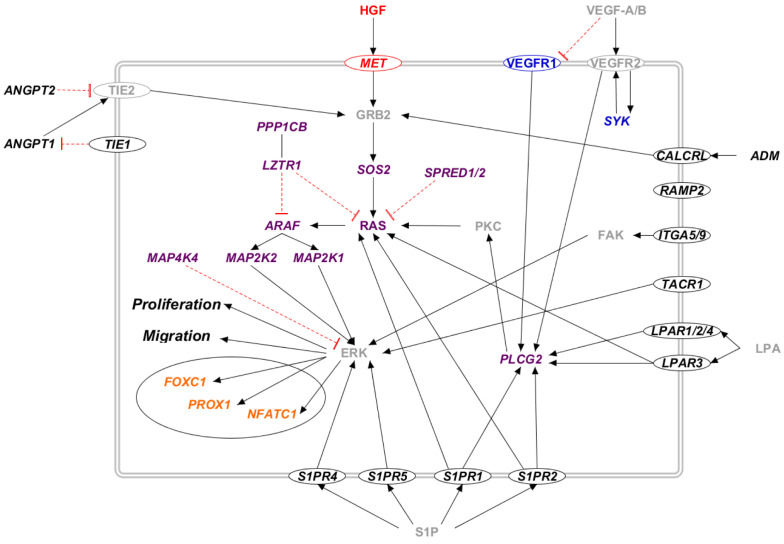
The RAS/MAPK pathway in candidate genes for primary lymphedema. Proteins that participate in one of the previously reported molecular pathway are colored: RAS/MAPK (violet), VEGF-C/VEGFR-3 (blue), and HGF/MET (red). The transcription factors are marked in orange. Proteins that do not participate in any of the previously reported pathways are marked in black. Proteins not encoded by either diagnostic or candidate genes are marked in grey. Proteins encoded by candidate genes are reported in italic. When circled, the protein is a membrane protein. Black arrows represent a positive interaction, while red T-bars represent an inhibition.

**Figure 5 ijms-23-07414-f005:**
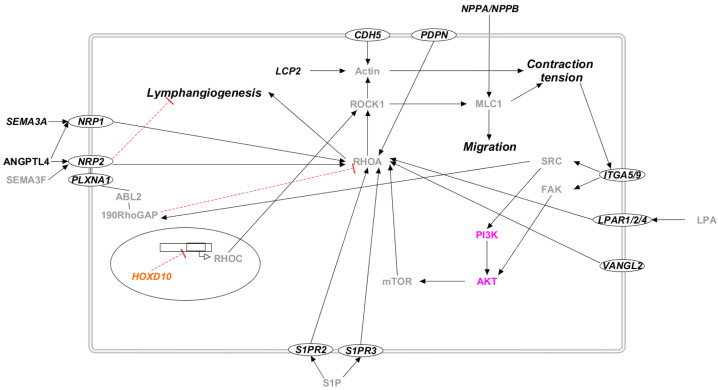
The Rho/ROCK pathway in candidate genes for primary lymphedema. Proteins that participate in PI3K/AKT molecular pathway are colored in lilac. The transcription factors are marked in orange. Proteins that do not participate in any of the previously reported pathways are marked in black. Proteins not encoded by either diagnostic or candidate genes are marked in grey. Proteins encoded by candidate genes are reported in italic. When circled, the protein is a membrane protein. Black arrows represent a positive interaction, while red T-bars represent an inhibition.

**Figure 6 ijms-23-07414-f006:**
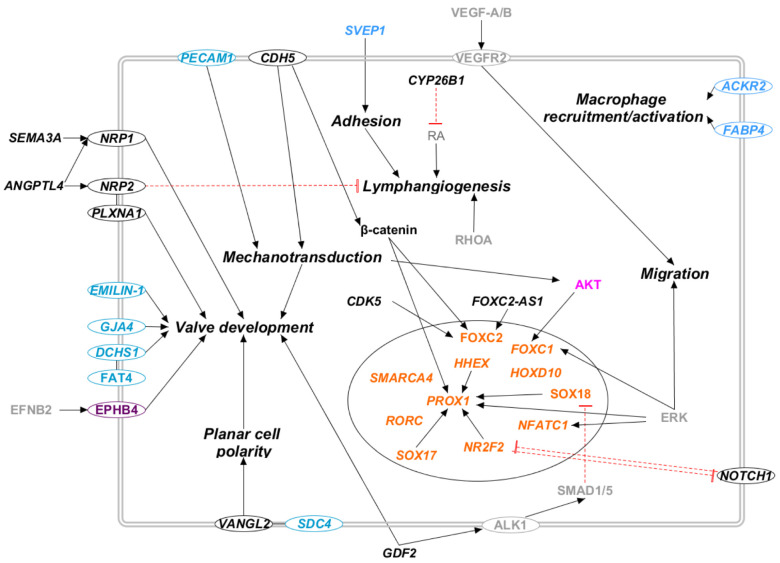
Secondary pathways in candidate genes for primary lymphedema. Proteins that participate in one of the previously reported molecular pathway are colored: RAS/MAPK (violet), PI3K/AKT (lilac). The transcription factors are marked in orange, and the light blue proteins are those whose outcome is known, i.e., altered lymphatic valve formation, but the molecular pathway is uncertain. Proteins that do not participate in any of the previously reported pathways are marked in black. Proteins not encoded by either diagnostic or candidate genes are marked in grey. Proteins encoded by candidate genes are reported in italic. When circled, the protein is a membrane protein. Black arrows represent a positive interaction, while red T-bars represent an inhibition.

**Table 1 ijms-23-07414-t001:** Characteristics of the probands analyzed for this study.

Characteristic		Case Subjects (*n* = 147)
Age	Mean	46 ± 18
	Median	47 ± 18
Females/Males		116/31 (79%/21%)
Period of onset	Congenital	7 (5%)
	Childhood (1–10 years)	26 (18%)
	Youth (11–17 years)	43 (29%)
	Adult (>18 years)	71 (48%)
Age of onset	Mean	27 ± 18
	Median	25 ± 18
	Unknown	*n* = 39
	Sporadic	79 (54%)
Familiarity	Familiar	40 (27%)
	Unknown	28 (19%)
Location	Lower limb	49 (33%)
	Lower limbs	77 (53%)
	Upper limb	6 (4%)
	Lower and upper limbs	12 (8%)

## Data Availability

Data are contained within the supplementary material.

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
