# Peer review of "Low Efficacy of Genetic Tests for the Diagnosis of Primary Lymphedema Prompts Novel Insights into the Underlying Molecular Pathways"

_ijms, 2022, doi:10.3390/ijms23137414_

Round 1
Reviewer 1 Report
This manuscript, based on the observation that an almost complete set of known lymphedema genes only explains 7% of a cohort of 147 individuals, provides very interesting lists of candidate genes shown in kind of functional manner subdivided in core signaling pathways. It will be helpful in the search of the still likely unknown causes of primary lymphedema. The novel genes that have been shown at the origin of the disease during the last years only account for a small percentage of the cases, showing the diversity of genetic alterations that can be at the base of a same disorder, clinically-speaking. The manuscript is globally well written, but requires some adjustments as follows :
- Point 2.1, Figure 1 and methods: From the text and methods, it is not completely clear that the list of genes that were screened corresponds to Supplementary Table 1. I kind of deduced it, but If it is the case, you should at least refer to Table 1 when saying “Figure 1 reports the results of the genetic tests in the probands recruited for this study.” so that it becomes clear that this is the list of genes that you explored. From the M&M, your refs 7 and 8 represent only 2 and 10 genes, respectively. On top of that, you should move the two first sentences of ‘M&M point 4.2’ into the section ‘4.1 Genetic analysis’ where it is more appropriate.
- It would also be interesting to cite the few genes that account for the 11 positive patients. I guess it includes FLT4, FOXC2 and a few that the authors previously reported. I do not expect that it will be 11 different genes …
- Along with the genetic results, the criteria that you used to determine whether a variant is considered inconclusive or Positive (thus a mutation) have to be detailed.
- The authors should state when (what year) their diagnostic list was established (and their panel designed) to understand why some of the genes more recently published including by the authors (ANGPT2, TIE1, …), or considered as confirmed lymphedema genes by others (e.g. Brouillard et al, Nat Rev Disease Primer 2021) (such as SOS2, RAF1, …), are not part of this diagnostic list but only considered as candidates ?
- In Figure 2, why are proteins like MET or IKBKG which are not considered as genes related to primary lymphedema but rather in Table S2, as well as proteins outside this list such as GRB2, MTOR or others, also colored ? In contrast, RIT1 and RASA1 are missing. Similarly, recognized lymphedema genes such as ANGPT2/TIE1 should also be shown here. Still in this figure: CELSR1 is also a membrane protein. It should therefore be circled and at the membrane.
- In Figure 3 and following ones : what are the proteins in Black? This is not not explained in the legend of Fig2 referred-to. Same question for the Green ones.
Not all black proteins are reported in Table S2. How to distinguish those that are int the Tables and those not ? Where does SEMA3F come from? You have SEMA3A listed in Table S2. Why genes not listed in supplementary tables are here in color, e.g. VEGF-A/B ?
- In Figure 5, why is NRP2 in black but NRP1 in blue ? RHOC, ROCK1 e.g. are not in supplementary tables but here in color.
- In the discussion, some of the references are not correct. Here are some examples: “First, heterozygous loss-of-function mutations to VEGF-C [33]”, Ref 33 is about PTPN11, not VEGFC;
“VEGF-C is expressed where the first lymph sacs develop during embryogenesis and in regions of lymph vessel sprouting [36].”, Ref 36 relates to PIK3CA somatic mutations, not VEGFC;
“Mosaic gain-of-function mutations [51] to AKT”, not sure this is the correct since ref. 51 is a review on RAS isoforms;
Next sentences: refs 52-54 are not related to PI3K but RAS proteins;
The reference (Bui & Hong, 2020) is not formatted.
Please check carefully that the references are properly assigned.
- Last paragraph of 3.1.3, isn’t it rather NRP and ANGPTL than NRP and ANGPT ? Also, do you have a reference for “Other candidate genes that participate in the PI3K/AKT path- way are ARAP3, SYK, RELN, and SVEP1.”
- What is the order of the genes listed in supplementary Table 1 ? Table 2 is alphabetical, but not this one.
- In Table S1, how come PTPN4 is activating ? It is a phosphatase of VEGFR3, so it should have the opposite effect …
- Well-established genes associated with lymphedema, such as TSC1, TSC2 were not considered by the authors whereas genes causing rather lymphatic-type anomalies (PIK3CA, AKT, …) are there. Why ?
Author Response
We thank the Reviewers of this manuscript for their very helpful comments that have clearly substantially contributed to the improvement of this manuscript.
Please see the attachment for the point-by-point rebuttal of all the issues raised by the Reviewers.
Thank you again and best regards,
Gabriele Bonetti

Reviewer 2 Report
I enjoyed reading the article by Gabriele Bonetti, Stefano Paolacci, Michele Samaja, Paolo Enrico Maltese, Sandro Michelini, Serena Michelini, Silvia Michelini, Maurizio Ricci, Marina Cestari, Astrit Dautaj, Maria Chiara Medori and Matteo Bertell titled: Low efficacy of genetic tests for the diagnosis of primary lymphedema prompts for novel insights into the underlying molecular pathways.
The authors took up the difficult topic of lymphedema diagnostics and its molecular mechanisms.
In my opinion, the work is very well composed. It has a rich visualization and a well-described discussion.
I think it will also be a valuable item due to the extensive literature review.
I define the scientific value of the article very highly.
Author Response

(The authors gave the same response as above.)

Round 2
Reviewer 1 Report
I went through the revised manuscript. The authors properly
addressed my concerns. The color-coding and legends of the figures are
also much more clear now. However, there are still references in
duplicate. To cite a few that catch my eyes: ref 90 is the same as 67;
ref 38 = 74, or ref 89 = 185. I have not gone through the entire
reference list one by one, thus, there might be other ones that I have
not seen. I let your editorial staff deal with this issue.